# A Real-Time Application for the Analysis of Multi-Purpose Vending Machines with Machine Learning

**DOI:** 10.3390/s23041935

**Published:** 2023-02-09

**Authors:** Yu Cao, Yudai Ikenoya, Takahiro Kawaguchi, Seiji Hashimoto, Takayuki Morino

**Affiliations:** 1Program of Intelligence and Control, Cluster of Electronics and Mechanical Engineering, School of Science and Technology, Gunma University, 1-5-1 Tenjin-cho, Kiryu 376-8515, Japan; 2SANDEN RETAIL SYSTEMS CORPORATION, Tokyo 101-8583, Japan

**Keywords:** smart vending machines, small data, k-means, convolutional neural network, visual interpretation, real-time application

## Abstract

With the development of mobile payment, the Internet of Things (IoT) and artificial intelligence (AI), smart vending machines, as a kind of unmanned retail, are moving towards a new future. However, the scarcity of data in vending machine scenarios is not conducive to the development of its unmanned services. This paper focuses on using machine learning on small data to detect the placement of the spiral rack indicated by the end of the spiral rack, which is the most crucial factor in causing a product potentially to get stuck in vending machines during the dispensation. To this end, we propose a k-means clustering-based method for splitting small data that is unevenly distributed both in number and in features due to real-world constraints and design a remarkably lightweight convolutional neural network (CNN) as a classifier model for the benefit of real-time application. Our proposal of data splitting along with the CNN is visually interpreted to be effective in that the trained model is robust enough to be unaffected by changes in products and reaches an accuracy of 100%. We also design a single-board computer-based handheld device and implement the trained model to demonstrate the feasibility of a real-time application.

## 1. Introduction

Since the earliest known vending machine was invented by Heron of Alexandria [1,2], the vending machine, a machine dedicated to functioning as an unmanned store for convenience to humans, has undergone a long history of development. The first commercial vending machines date back to the 1880s [1], and today, the vending machine is shifting from a single-function store with self-service to a station that can provide more diverse services [3]. The significance of vending machines in modern society is self-evident. Owing to its compact size, vending machines can be installed in many poor locations where other retail business forms are not suitable for installation, such as cramped corners or inaccessible suburbs, which exceedingly prolong the reach of the conventional retail distribution network [4,5,6]. The number of installed vending machines worldwide is remarkably large and it has affected people’s living habits in an insignificant way, making it likely that one buys drinks, snacks, or even daily groceries from a vending machine at some point in the day. There are numerous vending machine-related businesses in the U.S. and Japan [3,7]. It has commonly been assumed that the U.S. is the country with the most installations, and presently its vending machine business is a USD 31 billion industry [8]. However, in terms of the number of installations per person, Japan has the highest density of vending machines [3] with over 4 million vending machines nationwide by the end of 2021 [9]. The massive presence of vending machines, especially in various transportation institutions with large traffic such as airports and subway stations, can enhance the visibility of brands to strengthen brand cognition for marketers [6]. On the other hand, for consumers, 24-h instant transactions without intermediaries bring great convenience and save significant amounts of time [6].

As one of the most pervasive retail business forms, the vigorous development of the vending machine industry demands more people to join the business; to do this, the most crucial thing is the shopping experience of consumers. In terms of consumer experience, what makes them satisfied and what makes them complain has a tremendous impact on the development of the vending machine industry. Given that retail stores are the closest alternative to vending machines and compete with their core benefits [6], the differences in consumer experience between the two can reveal the drawbacks of vending machines. It is astonishing that inadequate refund and complaint processes, which are highly associated with self-service, one of the advantages of vending machines, are the main culprits of consumers’ unsatisfied experiences [6,10]. The vast majority of unsatisfactory experiences resulting in consumers wanting a refund or complaining come from products getting stuck in the machine [6,11]. It is crucial to ensure that vending machines succeed in dispensing products to help consumers make a smooth purchase and avoid consumers’ unsatisfied experiences [11,12,13]. The loss of consumers is a heavy blow to the growth of the vending machine industry, which we desire to prevent.

The importance of big data for machine learning is undeniable [14,15,16]. The accepted open source datasets for validating machine learning algorithms in vending machine scenarios are still severely lacking [17], which will slow down the development of the unmanned retail business. Collecting data and labeling them correctly is time-consuming and labor-intensive, and for some real-world situations, one has to deal with small data. In recent years, many researchers have attempted to apply machine learning to small data [18,19,20,21]. Despite the fact that there is much literature related to unmanned retail and smart vending machines, there is a complete lack of literature on dealing with the problem of products getting stuck to the best of our knowledge. It is also worth noting that the trade-off between the inference time and accuracy of machine learning algorithms has been considered for real-time applications [17,22,23,24].

In this work, we aim to detect the end of the spiral rack used to dispense products to consumers with machine learning. The dispensation mechanism of a spiral rack-mounted vending machine and its relationship with whether products are likely to get stuck will be explicitly described in the next section. We collected a total of 576 images of 18 different products evenly via a webcam which was labeled into four classes due to the fact that, to the best of our knowledge, there is no public and accepted dataset for solving the trouble of products being stuck in vending machine scenarios. Our machine learning-based algorithm was trained on such small data yet turned out to be highly accurate and requires low computational cost. Additionally, visual interpretation with the help of gradient-weighted class activation mapping (Grad-CAM) [25] ensures that our model has learned explainable and robust features. Furthermore, we implemented the trained model on a Raspberry Pi connected to a webcam and a liquid-crystal display (LCD) and verified the feasibility of the real-time application.

### Related Works

Rapid advances in mobile payment, IoT, and AI technologies in recent years have promoted the development of the unmanned retail business. Most effort for smart unmanned vending machines has been made on the detection and classification of objects in vending machine scenarios.

Zhang et al. [17] constructed a large-scale dataset of over 30,000 images comprising 10 categories of beverages and pointed out the lack of open source datasets optimized solely for object detection and classification in vending machine scenarios. Xu et al. [22] and Liu et al. [23] focused on solving the difficulties in object detection caused by the mutual occlusion of products in vending machines due to dense placement. They collected a dataset of 18,000 images containing 20 types of beverages by a binocular camera system, which was later expanded to 21,000 images [22,23]. However, this dataset has not been made public. Kim et al. [24] proposed a system for recognizing purchasing behavior by detecting and tracking products in real-time, where the system’s object detection model was trained on the PASCAL VOC0712 dataset [26] composed of approximately 20,000 images. Note that the above researchers relied on big data to develop their algorithms and they focus on the products inside the vending machine rather than the vending machine itself.

## 2. Problem Formulation

At present, there are at least the following types of racks used to dispense products to consumers in vending machines:Spiral rack: Products are placed between contiguous spirals. It is widely used as it fits different shapes of products;Conveyor rack: Products are placed on the conveyor. It is highly stable and suitable for box-shaped products;Hook-type rack: Products are hung on the hook. It requires products that have a placement hole on the top.

In this work, we focus on the spiral rack-mounted vending machines, as shown in Figure 1, for their wide suitability for various shapes of products.

### 2.1. Factors in Causing Stuck Products

From the experience of numerous trials, a few factors can contribute to the product being stuck. The width of the two partitions where spiral racks are placed in the middle limit the size of products that can be placed inside the spirals. A narrow width is highly probable to cause products exceeding a certain size to get stuck between the partitions during dispensation. The spiral pitch decides the maximum thickness of products, and products that are too thin usually fail to fall into the hands of consumers.

Taking into account the fact that the operator will follow the manual or check visually to place products in the partition width and spiral pitch appropriate for their sizes, the most critical factor for the product being stuck is the placement of the spiral rack, which can be indicated by the end of the spiral rack. The end of the spiral rack is defined as the last portion of the spiral rack closest to the consumer from the front view of the vending machine cabinet. When the end of the spiral rack is in a low position as illustrated in Figure 1b, the spiral pushing force, which comes from a motor that drives the spiral rack to rotate a full circle after payment, stops before the product completely passes through the partitions, resulting in the product being stuck. On the contrary, when the end is in a high position, as shown in Figure 1c, there is basically no problem of getting stuck.

### 2.2. Research Purpose

Although the relationship between the end of the spiral rack and whether the product is going to be stuck has obvious regularity, it is not absolute. From experience, the placement in Figure 1b is the most likely to cause the product to get stuck, but it is not stuck every time. When the end of the spiral rack is upside down, as shown in Figure 1c, products can basically fall into customers’ hands smoothly.

From the perspective of the vending machine operator, the habit of placing the spiral rack, taking the spiral rack on the right as an example, is named after the “Down”, “Left”, “Right” and “Up” as defined in Figure 2. We take the classification of the end of the spiral rack as the starting point for analyzing whether the product will get stuck, and by using machine learning, we design a four-class classifier model on small data and implement the model in a real-time application.

## 3. Dataset

In this section, to demonstrate the feasibility of our model on small data due to real-world limitations, we construct a dataset with 4 classes to classify, each of which equivalently contains 18 products with only 8 images per product. The way that products can be placed in a compact space is limited, resulting in the fact that even if the amount of data is increased, the features of the data are not increased. The collection will be briefly introduced in the following subsection. To demonstrate that our model is not affected by changes in products, we then use a k-means clustering-based method to split the collected dataset appropriately into a training dataset and a validation dataset with no product overlap.

### 3.1. Image Collection

Our data acquisition platform is based on a webcam connected to a computer, which is fixed at a distance of about 20 centimeters in front of the product shelf. All properties of the camera are fixed and images are captured directly as gray-scale images rather than color images. Notably, in order not to affect the overall image due to the change of the product, the auto-focus and auto-exposure functions are disabled, and instead, a constant focus and a constant exposure are manually tuned at which the end of the spiral rack is clearly visible. Because in actual operation, for example, in the automatic exposure mode, the brightness of the product will affect the brightness of the overall picture, causing the spiral to sometimes be clear and sometimes buried in shadows. Only the front of each product is used in consideration from the consumers’ perspectives. The products are translated, slightly rotated, and flipped left and right in the confined space of the spirals to make the eight images as distinguishable as possible.

Eventually, there are a total of 144 original images in each class with the size of 1280×720, covering 18 different products (including a situation where no product is placed). There is no specific selection of certain products, but rather as many products as we have on hand. Despite the fact that only snacks in rectangular packaging are collected, the collection will not solely be beneficial for models optimized for snacks because it is conceivable that grocery products such as tissues and face masks are no different in their rectangular packaging.

### 3.2. Data Processing and Proposed Splitting Method

The motivation for splitting data in an appropriate way is that biased data may cause the features learned by the model to be biased, that is, unable to cope with general situations. Imagine a situation where a model trained only on data from person A’s handwritten digits would perform poorly in predicting the recognition of person B’s handwritten digits, or more broadly anyone’s handwritten digits, especially when person A’s handwriting has strong personal characteristics. For such special training situations as the aforementioned example, it is reasonable to believe that no matter how much effort is put into the model, it is difficult to have the expected performance, that is, applicability in a broad sense. Therefore, it is essential to split the dataset appropriately.

Originally captured images of left and right spiral racks are center-cropped into the size of 420×420 to ensure that the spiral edge remains approximately $\frac{1}{8}$ of the image width or height from the image edge, and the images of the left spiral racks are flipped to the right to double the amount of data. It is a common method to split the whole dataset randomly into a training dataset and a validation dataset according to certain proportions. The random split enables the training dataset and the validation dataset to have feature similarity when there are sufficient data. However, this may not be applicable to small data where an extreme situation that the randomly split data for training cannot represent the whole dataset may occur. Therefore, in the case of small data, it is necessary to ensure that the training dataset is sufficient so that it can represent the whole dataset, and that the validation dataset is sufficient to validate the trained model on rich features. We assume that the changes in the features of the processed image mainly come from the changes in the image intensity and pattern complexity of the product itself, which can be roughly considered as the mean μ and standard deviation σ of the image, respectively. The μ and σ of the *n*th product are calculated from the average of all images belonging to the *n*th product, as denoted in Equations (Equation 1) and (Equation 2). The two features are then normalized to zero mean and unit variance, respectively, for convenience in k-means clustering.
(1)$$\mu_{n}=\frac{1}{N\times W\times H}\sum_{n=1}^{N}\sum_{i=1}^{W}\sum_{j=1}^{H}I_{n}\left(i,j\right)$$
(2)$$\sigma_{n}=\frac{1}{N}\sum_{n=1}^{N}\sqrt{\frac{1}{W\times H}\sum_{i=1}^{W}\sum_{j=1}^{H}{\left(I_{n}\left(i,j\right)-\mu_{n}\right)}^{2}}$$
where *W* and *H* denote the width and height of the processed image, respectively, and $I_{n}$ denotes the pixel intensity. *N* is the number of images for the *n*th product.

Let $x_{n}=\left(\mu_{n},\sigma_{n}\right)$ denote the feature vector of the *n*th product. Given a set of features $X=\{x_{n}|x_{n}\in R^{2},\;1\leq n\leq 18,\;n\in N\}$, we aim to partition the 18 (number of product types) features into *k* clusters $C=\{C_{1},C_{2},\ldots ,C_{k}\}$ so as to minimize the within-cluster sum of squares (WCSS) of each point to its cluster centroid $c_{i}\;\left(i=1,2,\ldots ,k\right)$, where Equation (Equation 3) defines the objective. One can refer to Lloyd’s algorithm [27] to obtain a certain local optimum for this problem in a simple and computation-friendly way.
(3)$$\underset{C}{\arg\min}\sum_{i=1}^{k}\sum_{x_{n}\in C_{i}}{∥x_{n}-c_{i}∥}^{2}$$
where $c_{i}$ is the mean vector of cluster $C_{i}$.

The appropriate *k* can be either selected by the elbow method which is by graphing the relationship between the number of clusters *k* and the WCSS, and picking the elbow of the curve as the optimal number of clusters to use [28], or by the silhouette method where the peak of the curve of *k* versus the average silhouette value indicates the optimal number of clusters [29]. For a point $x_{n}\in C_{i}$, the silhouette value is defined as Equation (Equation 4).
(4)$$s\left(x_{n}\right)=\left\{\begin{array}{cc} \frac{b\left(x_{n}\right)-a\left(x_{n}\right)}{max\{a\left(x_{n}\right),\;b\left(x_{n}\right)\}}, & |C_{i}|\;>\;1\\ 0, & |C_{i}|\;=\;1 \end{array}\right.$$
where $|C_{i}|$ is the number of points in cluster $C_{i}$, $a(x_{n})$ measures the similarity of $x_{n}$ to its own cluster $C_{i}$ by the average distance of $x_{n}$ from the rest of the points in the cluster, as denoted in Equation (Equation 5), and $b(x_{n})$ measures the dissimilarity of $x_{n}$ from points in the nearest cluster as denoted in Equation (Equation 6).
(5)$$a\left(x_{n}\right)=\frac{1}{|C_{i}|-1}\sum_{x_{m}\in C_{i},\;n\neq m}{∥x_{n}-x_{m}∥}^{2}$$
(6)$$b\left(x_{n}\right)=\min_{i\neq j}\frac{1}{|C_{j}|}\sum_{x_{m}\in C_{j}}{∥x_{n}-x_{m}∥}^{2}$$

We then split the clustered data into two parts from each cluster, the training dataset $T=\{T_{1},T_{2},\ldots ,T_{k}\}$ and the validation dataset $V=\{V_{1},V_{2},\ldots ,V_{k}\}$, based on silhouette values within each cluster. Feature $x_{n}\in R^{2}$, as mentioned above, is used to refer to the *n*th product, and selecting this feature is the selection of the overall dataset of the product. The silhouette value is a measure of the similarity of a point to its own cluster compared to other clusters [30], where a high value indicates a good match to its own cluster and a poor match to neighboring clusters. Let $C^{\prime }=\left\{C_{1}^{\prime },C_{2}^{\prime },\ldots ,C_{k}^{\prime }\right\}$ be the sorted set of clusters, where each cluster is sorted by the descent order of silhouette values within itself. Here, $n_{i}$ is the local sorted index of $x_{n}\in C_{i}^{\prime }$ and $f_{i}\left(\cdot \right)$ maps the relationship of the local sorted index $n_{i}$ and its global product index *n*, denoted as $f_{i}:n\to n_{i}$. For the training dataset $T_{i}$, we take a pair of points with the maximum and the minimum silhouette at the same time and take the next pair after stepping an interval towards the average silhouette. The rest is treated as the validation dataset. In this way, it ensures that the training dataset and the validation dataset share a close average silhouette. Our splitting method for the training dataset $T_{i}$ and validation dataset $V_{i}$ in cluster $C_{i}^{\prime }$ is explicitly described in Equations (Equation 7) and (Equation 8), respectively. In our method, it is always that $|T_{i}\left|\;\geq \;\right|V_{i}|$.
(7)$$T_{i}=\left\{\begin{array}{cc} \{x_{n}|f_{i}\left(n\right)=1\}, & |C_{i}|\;=\;2\\ \{x_{n}|1\leq f_{i}\left(n\right)\leq |C_{i}|,n\in N\}, & |C_{i}|\;\in \;\left\{2n+1|n\in N\right\}\\ \{x_{n}|1\leq f_{i}\left(n\right)\leq \frac{|C_{i}|}{2},f_{i}\left(n\right)=2m+1,m\in N\}\cup \\ \{x_{n}|\frac{|C_{i}|}{2}+1\leq f_{i}\left(n\right)\leq \left|C_{i}|,f_{i}\left(n\right)=2m,m\in N\right\}, & |C_{i}|\;\in \;\left\{2n|n\in \left\{N-\left\{1\right\}\right\}\right\} \end{array}\right.$$
(8)$$V_{i}=C_{i}-T_{i}$$

### 3.3. Data Splitting Method for Comparison

In this section, a data splitting method contrary to our proposal is described and used as a comparison in Section 5. The stand of our proposal is that the data contains the features of the entire dataset as much as possible and is uniformly scattered, so its opposite is that the data are biased towards a single feature.

Therefore, the selection for the training dataset $T_{i}$ gives priority to the data of a single cluster $C_{i}$, and when the number of data in $C_{i}$ is not sufficient (as in our proposal, the number is 10), the data closest to cluster $C_{i}$ are selected. If the number in cluster $C_{i}$ is greater than or equal to 10, the data with the highest 10 similarities in cluster $C_{i}$ are selected. The similarity within a cluster can be measured by Equation (Equation 5), and Equation (Equation 9) can be used to indicate the similarity of point $x_{n}\notin C_{i}$ to cluster $C_{i}$, where a smaller value means more similar. Here, we only discuss the situation that $|C_{i}|\;<\;10$, since this is our actual situation.
(9)$$d_{i}\left(x_{n}\right)=\frac{1}{|C_{i}|}\sum_{x_{m}\in C_{i}}{∥x_{n}-x_{m}∥}^{2}$$

Let $D_{i}$ denote the sorted set of $x_{n}\notin C_{i}$ in ascending order of their similarity to cluster $C_{i}$, the bijection relationship of the local sorted index $n_{i}$ in $D_{i}$ and its global product index *n* is indicated by $g_{i}:n\to n_{i}$. The training data biased towards cluster $C_{i}$ and its validation data can described as Equations (Equation 10) and (Equation 11).
(10)$$T_{i}=\{C_{i},\{x_{n}|\;1\leq g_{i}\left(n\right)\leq 10-|C_{i}|,x_{n}\in D_{i},n\in N\}\}$$
(11)$$V_{c}=C-T_{i}$$

## 4. Network Architecture and Training

### 4.1. Network Architecture

Our network (Figure 3) is a 10-layer residual network mainly inspired by residual networks (ResNets) [31], where most convolutional layers have 3×3 filters and downsampling is performed by convolutional layers with a stride of 2. Pre-trained networks have been found to respond strongly to low-level features such as edges and blobs in their first convolutional layer [32,33,34], which along with the geometric characteristics of the spiral itself, inspire us to use a residual network that passes low-level features to later layers. Notably, the first convolutional layer has only 16 filters of size 5×5×1, so the numbers of filters in the subsequent layers are only 32, 64, and 128. Batch normalization (BN) is used after each convolution and before rectified linear unit (ReLU) activation. Our network ends with a global average pooling (GAP) layer and a fully connected (FC) layer, followed by a softmax layer with four classes. Additionally, two types of shortcut connections are inserted, where a solid line denotes an identity shortcut and a dotted line denotes that a 1 × 1 convolution is applied to match the increased dimensions. The architecture is shown in Figure 3 along with the inference process. See Table 1 for a detailed architecture.

The number of parameters our network has and the floating point operations (FLOPs) of the model are summarized in Table 2, which shows that our network is remarkably lightweight and computation-friendly.

### 4.2. Implementation Details

The image is down-scaled to 160×160 from the processed one mentioned in Section 3.2 and normalized based on the mean and standard deviation of the training dataset. We initialized the weights as in [35] and trained the network from scratch. We use stochastic gradient descent with momentum (SGDM) of 0.9 and a weight decay of 0.0001 to train the network. We used a mini-batch of size 64 because, in practice, a smaller mini-batch makes the training unstable after convergence, and a larger mini-batch is not suitable for small data. We use an initial learning rate of 0.1 that is decayed by a factor of 10 at 160 and 240 iterations, and terminate training at 320 iterations. Eventually, we pick the model with the lowest validation loss. We use data augmentation such as horizontal and vertical pixel translation, and random erasing [36] for training. For validating, we only evaluate the fixed view of the resized 160×160 image.

## 5. Results

### 5.1. Data Split Results

Although neither method of selecting the ideal *k* achieves the ideal curve in our case, k=5 is a promising option, as indicated in Figure 4. The corresponding clustering result with boundaries for k=5 is also illustrated in Figure 4. We also compare the cases of *k* from 2 to 5 and find that cluster 1 when k=3 has been broken down into sub-clusters for k=4,5 and thus forms clusters of more uniform size for k=5. Taking the above considerations into account, we select k=5 as the number of clusters.

The result of the training dataset and the validation dataset using the proposed splitting method is shown in Figure 5. The training dataset and the validation dataset that share a close centroid are evenly scattered on the overall dataset, and the average silhouette of the two is almost the same.

As mentioned in Section 3.3, several cases are prepared as comparison objects, as shown in Figure 6. Due to the scarcity of data, a dataset that is completely biased to a certain cluster does not exist, and instead, a dataset that is biased to a few clusters is used. From top to bottom, there are the results of the bias direction from cluster 1 to cluster 5, and for simplicity, the case of cluster 2 is omitted due to its extreme similarity to cluster 1. The case of cluster 2 only shifts the training data to the left by one data point. Due to the concentration of features, the centroids of the two datasets are far apart, and the distribution of the silhouette value is not uniform. The worst case is as illustrated in Figure 6b where there is no intersection of clusters.

### 5.2. Classification Results and Visual Interpretations

To evaluate the performances of models trained with cases discussed in Section 5.1 under the premise of using our proposed model, we compute the precision, recall, and the harmonic mean f1-score of precision and recall for each class, as defined, respectively, in Equations (Equation 12)–(Equation 14). Precision is the proportion of relevant instances among the retrieved instances, while recall can be thought of as a model’s ability to find all relevant instances of a class within a dataset [37]. The performance metrics of each trained model are summarized in Table 3, Table 4, Table 5, Table 6 and Table 7. Since the overall dataset is balanced in each class, it is safe to assess both models from a macro perspective, where the averages for all classes are computed and shown in Table 8.
(12)$$Precision=\frac{TP}{TP+FP}\times 100\%$$
(13)$$Recall=\frac{TP}{TP+FN}\times 100\%$$
(14)$$F1-Score=2\times \frac{Precision\times Recall}{Precision+Recall}\times 100\%$$
where *TP*, *FP* and *FN* stand for true positive, false positive, and false negative, respectively.

It is indisputable that our proposal of data splitting outperforms other cases and has no instances of wrong predictions. Among the comparison objects, case (a) and especially case (b) perform relatively poorly. It can be seen that since the number of cluster 2 dominates, cases (a) and (b) will be more biased than cases (c) and (d), causing them to perform poorly on the validation data. From the comparison, this indicates that our model is capable of achieving good performance even when there is a certain degree of bias in data splitting.

In recent years, more attention has been paid to the interpretation of models. We use Grad-CAM [25] to explain what features the models have learned to give an explicit understanding of correct/incorrect predictions. Randomly chosen images are predicted by our proposal and other cases, as shown in Figure 7.

The model using our proposed data splitting shows that it learns consistent features that focus on the end of the spiral rack for each class, thus avoiding the influence of the product’s size and pattern. At the same time, it also shows that the training did not lead to over-fitting, which is prone to occur on small data.

On the other hand, models as comparisons fail to learn the end of the spiral rack as features for each class. For cases (a) and (b), they show inconsistent prediction reasons for the same class, which indicates that different products affect the models’ predictions. For cases (c) and (d), although their accuracy is higher, the visual interpretation is not as good. In general, our proposed model performs well even when the data splitting is slightly biased. After using the proposed splitting method, not only does the accuracy reach 100%, but also with the help of Grad-CAM, the learned features are as expected.

### 5.3. Real-Time Implementation

To test the feasibility of our trained model and its computation cost in real time, we construct a system consisting of a Raspberry Pi, a webcam, and an LCD. The system has realized the following functions sequentially.

Capture: The webcam captures a front-view image in the way we collected it, as described in Section 3.1, and outputs to the Raspberry Pi;Image Processing: The captured image is cropped and resized to the input size of the trained model, as described in Section 3.2, on the Raspberry Pi side;Prediction: After image processing, the Raspberry Pi runs the trained model to obtain the prediction.Display: The predicted result is displayed on the LCD in real time once the prediction is obtained.

The specifications of the hardware are summarized in Table 9. Benefiting from the small number and small size, our system can support handheld if needed. The Raspberry Pi used is a small-scaled single-board computer with 8GB memory. The operating system is installed on a Micro SD card in advance. The LCD is a 16-character × 2-line display along with a module that converts the parallel signals interface into the inter-integrated circuit (I2C). The webcam is only for capturing images, and it can be replaced as long as the capturing conditions such as fixed focus and fixed exposure in the experiment are met.

Moreover, the connection between each component is illustrated in Figure 8. We power the Raspberry Pi 4 Model B with the official Raspberry Pi Power Supply via USB-C that outputs a 5.1v voltage. The webcam is directly connected to Raspberry Pi through USB. The operating voltage of the LCD and its adapter module is 5v, which can be supplied by the Raspberry Pi.

In the experimental setup, the trained model is first converted to the open neural network exchange (ONNX) format, which enables various deep learning frameworks to store network data and interact in the same format. The system is constructed as in Figure 9 and follows the above operation flow. A product is placed and then the LCD shows that the prediction is “*Up*”, which is consistent with the actual situation. Herein, we focus on the inference time that the proposed network itself costs, and thus the time spent in prediction, the third operation, is considered the computation cost of the network in real-time, as summarized in Table 10. Our proposed network is computationally light as expected, making it ideal for real-time use. In practice, the end of the spiral rack when the 18 different products were sequentially placed was correctly classified with confirmation by the display.

## 6. Discussion

Despite the achievements we made, we are yet to relate the relationship between the end of the spiral rack and the dispensation failure of specific products. Moreover, the camera used in this paper was placed in a fixed location for controlling experimental variations. In further work, we plan to explore a combination of object detection algorithms for products and spirals with our classification algorithm in this paper to remove the restrictions of camera placement and image cropping. There are two main options. One is to train two models in order to realize product detection and spiral classification, respectively, and the other is to integrate them into one model. The inference time and accuracy trade-off should still be our first consideration.

## 7. Conclusions

In this paper, we designed and constructed a real-time application with machine learning to detect the end of the spiral rack in the vending machine scenario. We first introduced a k-means clustering-based method to split small data due to real-world limitations into the training dataset and validation dataset. Our splitting method ensures that both datasets are similar in features and uniform in data number because, in order to verify whether the trained model is over-fitting, it is required that there is no product overlap between the two datasets and that the validation dataset has sufficient features to prove this instead of blindly pursuing more data for training. Compared to an extreme case that random splitting could cause, our proposal is promising to work even with products other than the 18 types used in this paper, since it helps the CNN model learn consistent features to ignore the product variations.

We then proposed a 10-layer residual network as the classifier model of four classes, where we focus on the lightweight of the model. In the case of small data and few features to classify, it is more suitable to use a simple and shallow network because there are more parameters to train for a deep network, which is contrary to the characteristics of small data. Deep networks have a higher risk of over-fitting at this time. Compared to a plain network that has no shortcut connections, we hope that with the help of residual learning, low-level features learned in early layers can be passed to sequencing layers. It is verified that our network not only has a small computational cost in theory but also in real-time application.

## Figures and Tables

**Figure 1 sensors-23-01935-f001:**
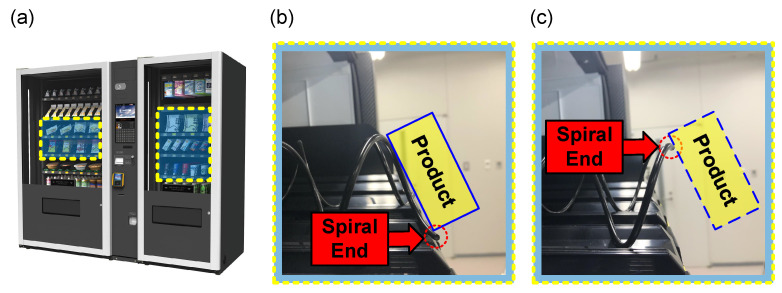
(**a**) An overall view of a multi-purpose vending machine in which most racks are spirals. The mechanism of dispensing products by the spiral rack with (**b**) a potential failure case; and (**c**) a successful case.

**Figure 2 sensors-23-01935-f002:**
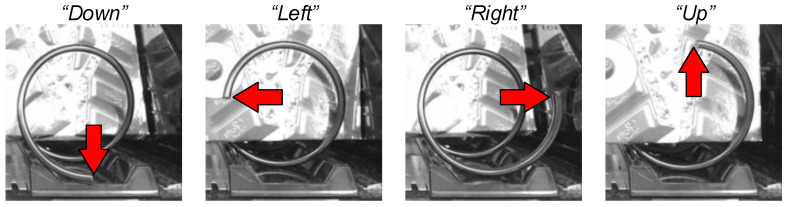
An example of the processed image, and how the 4 classes are defined by the direction the red arrow points to.

**Figure 3 sensors-23-01935-f003:**
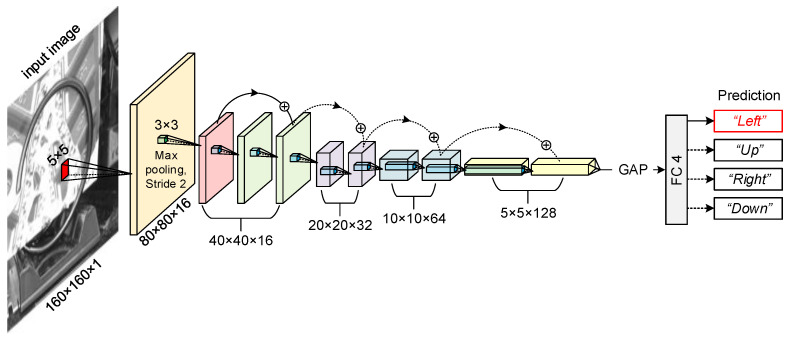
Network architecture with feature maps of sizes {80, 40, 20, 10, 5}, and an example of the inference process. The image is labeled as “Left” and predicted as “Left”.

**Figure 4 sensors-23-01935-f004:**
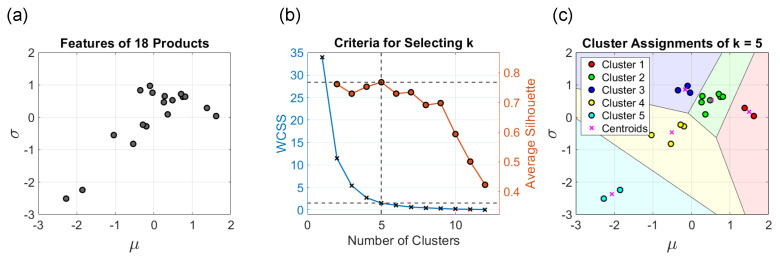
(**a**) The 2–D features where each point represents a type of product; (**b**) From the perspectives of both WCSS and silhouette, it is indicated that *k* is equal to 5; and (**c**) The clustering result when *k* is equal to 5.

**Figure 5 sensors-23-01935-f005:**
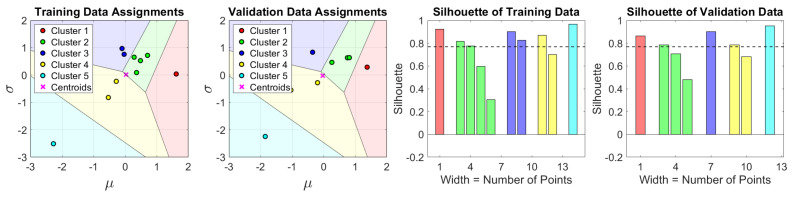
The split result of our proposal. The dotted lines in the right two plots show the average silhouette values for the corresponding left two plots.

**Figure 6 sensors-23-01935-f006:**
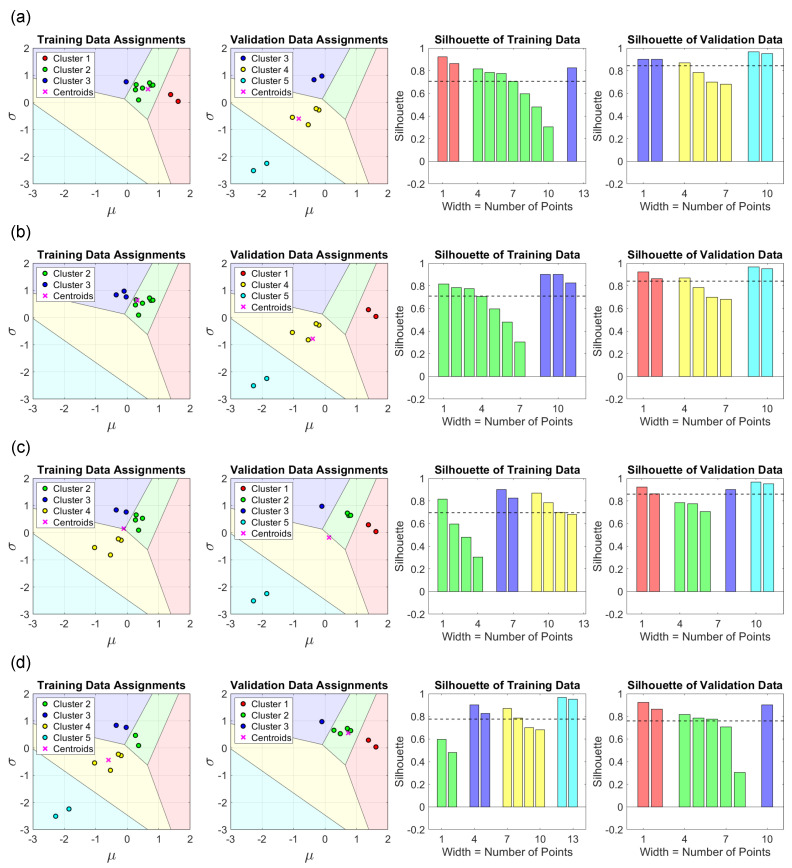
From top to bottom are the split results of the training data being biased towards (**a**) cluster 1; (**b**) cluster 3; (**c**) cluster 4; and (**d**) cluster 5, while the case of cluster 2 is omitted due to its similarity to cluster 1, as discussed in Section 3.3. The dotted lines in the right two plots show the average silhouette values for the corresponding left two plots. The worst case is shown in (**b**), where there is no intersection of clusters and the silhouette distributions are far apart.

**Figure 7 sensors-23-01935-f007:**
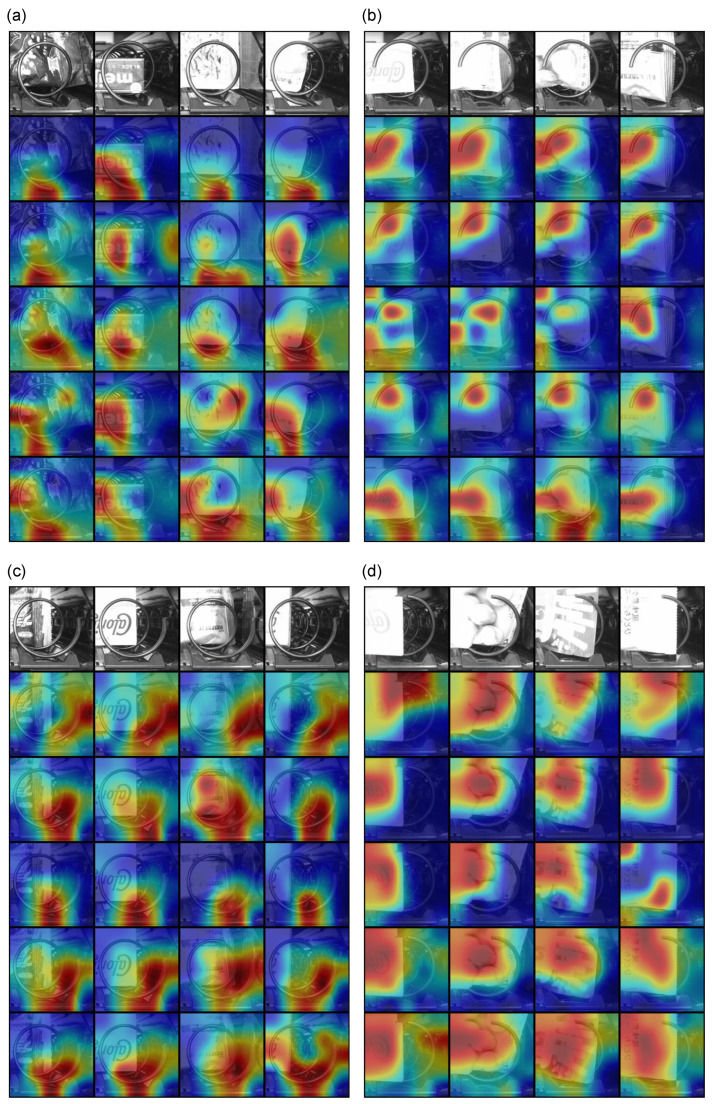
Grad-CAMs for 4 classes (**a**) “Down”, (**b**) “Left”, (**c**) “Right”, and (**d**) “Up”. For each subplot, the input images are in the first row, the second row shows the results of our proposal, and the subsequent rows show the results of cases (**a**–**d**).

**Figure 8 sensors-23-01935-f008:**
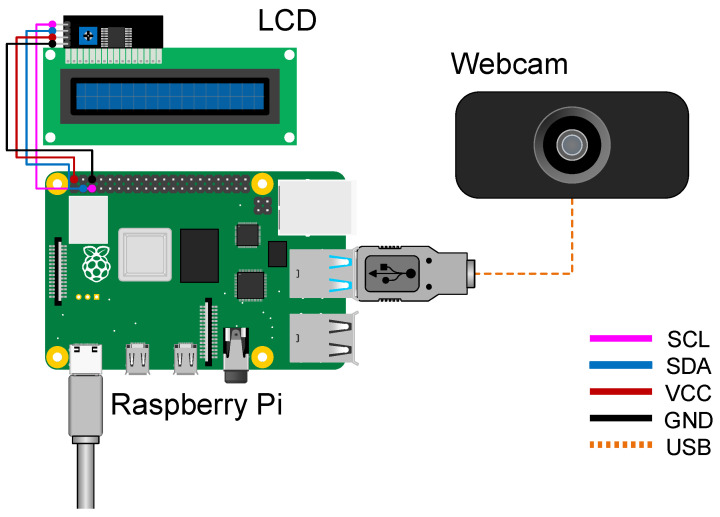
Design block diagram.

**Figure 9 sensors-23-01935-f009:**
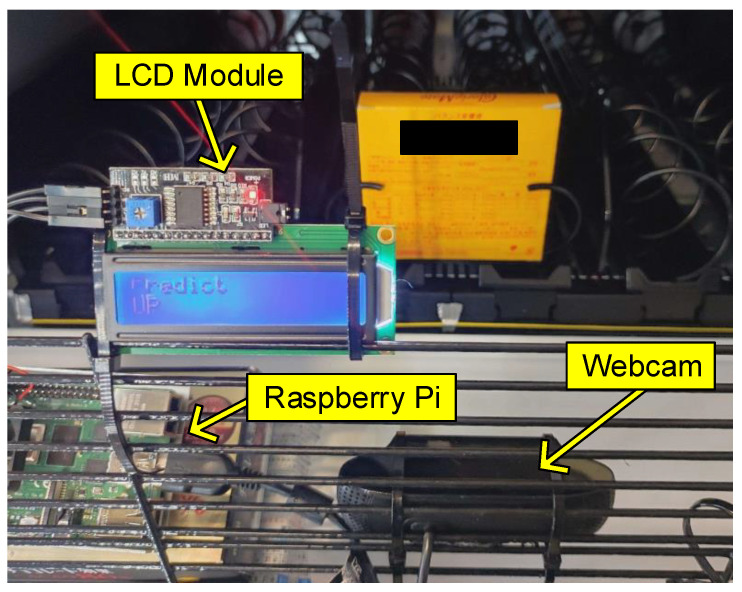
Experimental setup of our handheld device for real-time inference and display.

**Table 1 sensors-23-01935-t001:** Architecture of our proposed network. Blocks shown in brackets of the same convolutional layer are represented by the same color as in Figure 3. For instance, “3×3, 16” indicates 16 filters with a size of 3×3.

Layer Name	Output Size	Proposed Network
conv1	80×80	5×5, 16, stride 2
conv2	40×40	3×3, max pooling, stride 2
		$\left[\begin{array}{c} 3\times 3,16\\ 3\times 3,16 \end{array}\right]$, stride 2
conv3	20×20	$\left[\begin{array}{c} 3\times 3,32\\ 3\times 3,32 \end{array}\right]$, stride 2
conv4	10×10	$\left[\begin{array}{c} 3\times 3,64\\ 3\times 3,64 \end{array}\right]$, stride 2
conv5	5×5	$\left[\begin{array}{c} 3\times 3,128\\ 3\times 3,128 \end{array}\right]$, stride 2
classification	1×1	GAP, 4-d FC, softmax

**Table 2 sensors-23-01935-t002:** Computation cost of our proposed network.

Parameters	FLOPs
308 $\times \;10^{3}$	27.1 $\times \;10^{6}$

**Table 3 sensors-23-01935-t003:** Performance metrics of the network trained and validated with data using our proposed splitting method (Figure 5). There are no wrong predictions.

Class	*TP*	*FP*	*FN*	*Precision*	*Recall*	*F1-Score*
“Down”	128	0	0	100.00%	100.00%	100.00%
“Left”	128	0	0	100.00%	100.00%	100.00%
“Right”	128	0	0	100.00%	100.00%	100.00%
“Up”	128	0	0	100.00%	100.00%	100.00%

**Table 4 sensors-23-01935-t004:** Performance metrics of the network trained and validated with data in Figure 6a.

Class	*TP*	*FP*	*FN*	*Precision*	*Recall*	*F1-Score*
“Down”	128	68	0	65.31%	100.00%	79.01%
“Left”	94	32	34	74.60%	73.44%	74.02%
“Right”	115	0	13	100.00%	89.84%	94.65%
“Up”	75	0	53	100.00%	58.59%	73.89%

**Table 5 sensors-23-01935-t005:** Performance metrics of the network trained and validated with data in Figure 6b.

Class	*TP*	*FP*	*FN*	*Precision*	*Recall*	*F1-Score*
“Down”	128	150	0	46.04%	100.00%	63.05%
“Left”	14	38	114	26.92%	10.94%	15.56%
“Right”	124	27	4	82.12%	96.88%	88.89%
“Up”	31	0	97	100.00%	24.22%	38.99%

**Table 6 sensors-23-01935-t006:** Performance metrics of the network trained and validated with data in Figure 6c.

Class	*TP*	*FP*	*FN*	*Precision*	*Recall*	*F1-Score*
“Down”	128	0	0	100.00%	100.00%	100.00%
“Left”	128	0	0	100.00%	100.00%	100.00%
“Right”	128	8	0	94.12%	100.00%	96.77%
“Up”	120	0	8	100.00%	93.75%	96.77%

**Table 7 sensors-23-01935-t007:** Performance metrics of the network trained and validated with data in Figure 6d.

Class	*TP*	*FP*	*FN*	*Precision*	*Recall*	*F1-Score*
“Down”	128	0	0	100.00%	100.00%	100.00%
“Left”	128	0	0	100.00%	100.00%	100.00%
“Right”	128	2	0	98.46%	100.00%	99.22%
“Up”	126	0	2	100.00%	98.44%	99.22%

**Table 8 sensors-23-01935-t008:** Performance metrics comparison of the two models from a macro perspective. The model name indicates how the dataset is split.

Model	Accuracy	Macro Precision	Macro Recall	Macro F1-Score
Our proposal	100.00%	100.00%	100.00%	100.00%
Case (a)	80.47%	84.98%	80.47%	80.39%
Case (b)	58.01%	63.77%	58.01%	51.62%
Case (c)	98.44%	98.53%	98.44%	98.44%
Case (d)	99.61%	99.61%	99.61%	99.61%

**Table 9 sensors-23-01935-t009:** The handheld device configuration.

Webcam	LCD	Single-Board Computer	Operating System
Logicool C920n PRO HD	WayinTop 1602	Raspberry Pi 4 Model B	Ubuntu 20.04

**Table 10 sensors-23-01935-t010:** Computation cost in the real-time application, where the inference time is an average of 100 inferences.

File Size	Inference Time
1.192 MB	13.296 ms

## Data Availability

The data that support the findings of this study are available from the author, Seiji Hashimoto, upon reasonable request.

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
