# Peer review of "A Real-Time Application for the Analysis of Multi-Purpose Vending Machines with Machine Learning"

_sensors, 2023, doi:10.3390/s23041935_

Round 1
Reviewer 1 Report
The authors present a real-time application for the analysis of vending machines using machine learning.
However, the work falls far short of the proposed objective, as it is at a very early stage and with a lot of work to be done:
- The problem is not adequately explained. Both the abstract and the introduction suggest that that the problem of predicting jamming in this type of machine will be addressed, but in the end only a convolutional neural network is used to detect the position of the end of the spiral in which the products are housed.
- What relationship there is between the position of the end of the spiral and the possible jamming of the vending machine is not indicated.
- The neural network model used is not described in detail, nor are alternatives or other previously developed solutions discussed.
- The suitability of using machine learning and big data methods for problems with small data sets is neither justified nor demonstrated. This could be interesting, but the authors do not justify that this solution is effective.
- The experimentation is poor, and the reduced dataset does not demonstrate the feasibility of the proposed model.
- The implementation of the model on a small platform (Raspberry Pi) is interesting, but it is not described in detail or analysed in depth.
Reviewer 2 Report
The manuscript is presented nicely. However, please explain the impact of the Equations (Equations 1-8, 9-11) presented in the paper and the results. Why not the equations (Equations 9-11) presented in the methodology section?
Reviewer 3 Report
The authors move interesting tremeatics using machine learning to be used for wending machines. The authors present an interesting Introduction, in which they also de facto show content related to related work of other teams. In my opinion, it should be separated. A separately should be Introduction and Related Works separately. It would be certain that the article structure benefited.
The presentation of the problem and research was well presented and documented with appropriate mathematical expressions.
The authors document the received results well to show them in an attractive way and refer to them by providing them with relevant comments of the authors.
Technical attention: The same precision should be used in tables 3 and 4, so instead of 1.0 it should be 1,0000.
The authors skip the Conclusions chapter. In fact, Discussions they posted is more Conlusions. This should be separated and supplemented with the appropriate content that should be found in Conlusions and Discussion.
Round 2
Reviewer 1 Report
The authors present a real-time application for the analysis of vending machines using machine learning. They have made an effort to solve the issues listed, but there remains one problem that has not been adequately resolved:
- The experimentation is poor, and the reduced dataset does not demonstrate the feasibility of the proposed model.
Reviewer 3 Report
The authors responded to my suggestions and questions. Thank you.
